# Predicting terrorist attacks in the United States using localized news data

**Steven J. Krieg**[1], **Christian W. Smith**[2], **Rusha Chatterjee**[2], **Nitesh V. Chawla**[1] *

**1** Lucy Family Institute for Data and Society, University of Notre Dame, Notre Dame, IN, United States of America, **2** Physical Sciences Inc., Andover, MA, United States of America

* nchawla@nd.edu

**Data Availability Statement:** All data used in this study is publicly available from GDELT (https://www.gdeltproject.org/) and the GTD (https://www.start.umd.edu/gtd/).

## Abstract

Terrorism is a major problem worldwide, causing thousands of fatalities and billions of dollars in damage every year. To address this threat, we propose a novel feature representation method and evaluate machine learning models that learn from localized news data in order to predict whether a terrorist attack will occur on a given calendar date and in a given state. The best model (a Random Forest aided by a novel variable-length moving average method) achieved area under the receiver operating characteristic (AUROC) of $\geq 0.667$ (statistically significant w.r.t. random guessing with $p \leq .0001$) on four of the five states that were impacted most by terrorism between 2015 and 2018. These results demonstrate that treating terrorism as a set of independent events, rather than as a continuous process, is a fruitful approach—especially when historical events are sparse and dissimilar—and that large-scale news data contains information that is useful for terrorism prediction. Our analysis also suggests that predictive models should be localized (i.e., state models should be independently designed, trained, and evaluated) and that the characteristics of individual attacks (e.g., responsible group or weapon type) were not correlated with prediction success. These contributions provide a foundation for the use of machine learning in efforts against terrorism in the United States and beyond.

## Introduction

Terrorism poses a significant threat to society around the world. According to the Institute for Economics and Peace (IEP), in 2021 alone, global incidents of terrorism killed 7,142 people and caused tens of billions of dollars in economic damage [1]. Much of this activity occurs in the Middle East and other parts of Asia, leading many researchers to model the evolution and spread of terrorism in such regions. However, other areas of the world—including the United States—are not exempt from the effects of terrorism. Between 2015 and 2018, the Global Terrorism Database (GTD) recorded 229 incidents of terrorism in the United States [2]. The IEP also recently noted that the United States experienced one of the largest decreases in Positive Peace Index, a change that indicates increasing social disorder and greater risk of politically-motivated violence [3]. Predicting such violence before it occurs could enable prevention strategies and save lives.

**Funding:** This material is based upon work supported by the Army Contracting Command - Aberdeen Proving Ground, Edgewood Division under Contract No. W911SR-19-C-0007, which supported S.J.K., C.W.S., and R.C. The sponsors played no role in the study design, data collection and analysis, decision to publish, or preparation of the manuscript.

**Competing interests:** The authors have declared that no competing interests exist.

Several prior works address the problem of detecting terrorist activity, such as classifying pro-terrorism tweets [4]. However, in this study, we are interested in the problem of prediction rather than detection. Additionally, our task is to predict the occurrence of attacks or events, which is distinct from works that have sought to infer characteristics of an attack, such as the responsible group, after it has taken place [5–9]. Few works have attempted to predictively model terrorist attacks, with the recent work of Python et al. being one exception [10]. The authors trained several machine learning models using prior terrorism data (from the GTD) in conjunction with other geographic and socioeconomic features to predict attacks at discrete spatiotemporal intervals. Despite promising results and thorough analysis at an impressive and global scale, we found two fundamental problems with their approach:

1. Coarsely evaluating a model that represents a large region (e.g., West Africa) but comprises many smaller geographic cells overlooks the imbalanced spatial distribution of attacks. In this case, which is more complex than evaluation on a typical class-imbalanced dataset, measures like area under the receiver operating characteristic curve (AUROC) and area under the precision-recall curve (AUPRC) can be misleading. For example, using the data described in Table 1 and Fig 1, we can create a terrorism model for the United States that predicts an attack will occur every day in five states (CA, NY, TX, FL, and WA) but that no attacks will occur in other locations. If evaluated coarsely across the entire United States, this model would produce an AUROC of 0.733 and AUPRC of 0.468, both of which significantly outperform their respective baselines of 0.500 and 0.003. However, these outputs provide no value beyond reflecting that some regions have experienced a greater number of attacks. We avoid this problem by evaluating predictive performance for each region (state) separately.

2. In order to make meaningful predictions at a granular timescale (e.g., will a terrorist attack take place during a given week), models must have inputs of a similar temporal granularity in order to differentiate between points in time. Many of the inputs utilized by Python et al. in [10] are relatively static, such as population density and gross domestic product (GDP). The only temporally granular inputs were autoregressions based on local terrorist activity (i.e., whether there were recent terrorist attacks in the location of concern). While these autoregressions are valuable in their ability to model continuous terrorist activity, without

**Table 1. Terrorist attack counts between Feb. 18, 2015, and Dec. 31, 2018.**

| State(s) | # Attacks | Imbalance |
|---|---|---|
| CA, NY | 24 | .0170 |
| TX | 18 | .0127 |
| FL | 17 | .0120 |
| WA | 14 | .0099 |
| LA, MO | 7 | .0050 |
| NV, PA | 6 | .0042 |
| IN, NC, TN, VA | 5 | .0035 |
| CO, IA, MS, NM | 4 | .0028 |
| GA, IL, KY, MA, MN, ND, OH, OR | 3 | .0021 |
| AZ, DC, MD, MI, NE, NJ, SC, UT, WI | 2 | .0014 |
| CT, DE, ID, KS, MT, WY | 1 | .0007 |
| AK, AL, AR, HI, ME, NH, OK, RI, SD, VT, WV | 0 | — |

Imbalance is the class imbalance ratio (i.e., the proportion of the 1,413 days on which attacks occurred).

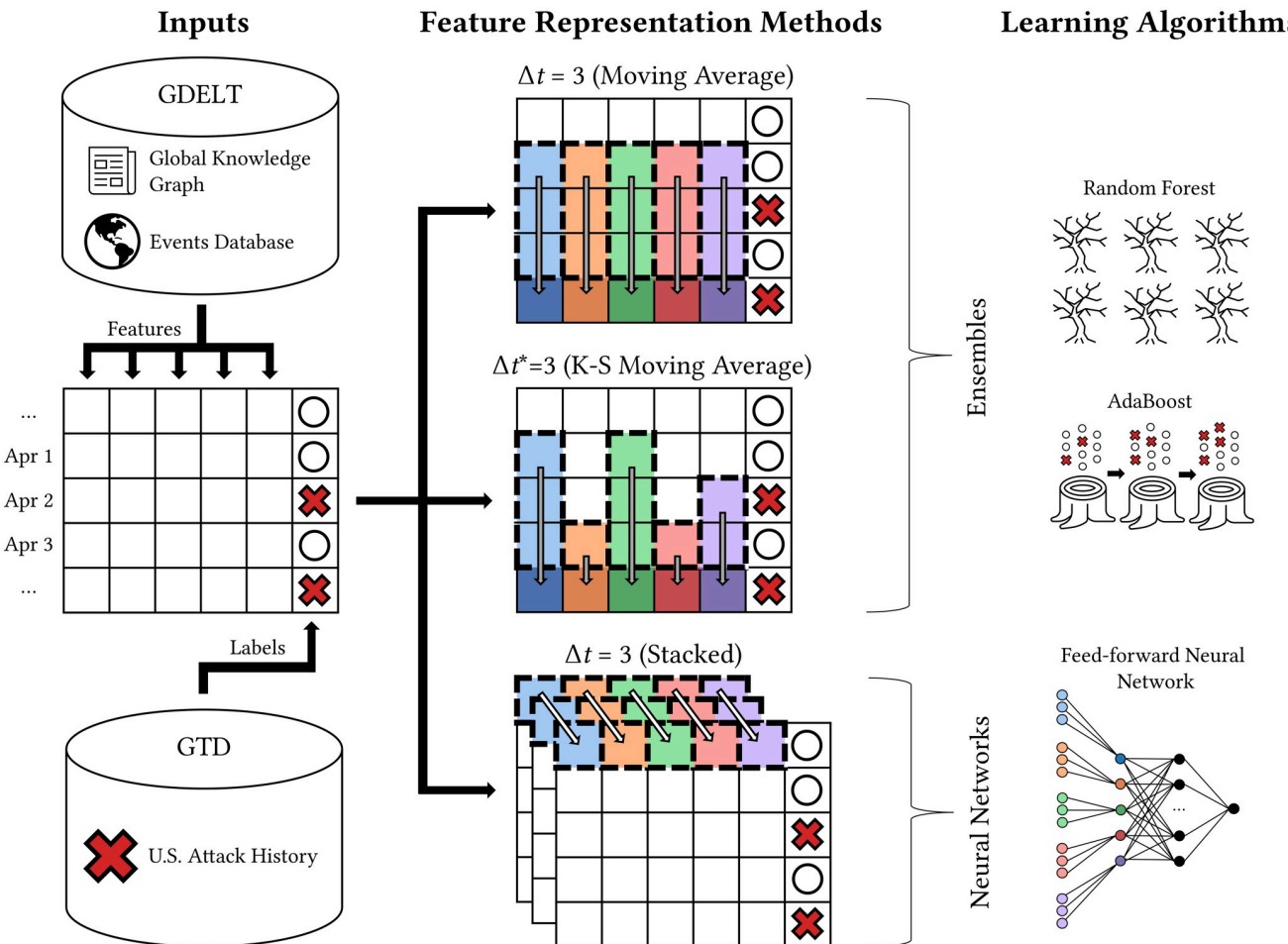

**Fig 1. Locations of the 229 terrorist attacks perpetrated in the United States between Feb. 18, 2015, and Dec. 31, 2018, as recorded in the Global Terrorism Database.**

other temporally relevant inputs the predictive models will be fundamentally limited in their ability to predict attacks at new or unusual locations. This shortcoming could also exacerbate the coarse evaluation problem described above.

Buffa et al. also frame terrorism as a binary classification problem, using a number of geospatial and population features to predict the location of terrorist attacks in Europe [11]. However, with respect to future predictions, their methods suffer from the same problems described above.

In contrast to the treatment of terrorism as a classification problem, a number of prior studies have modeled terrorism as a spatiotemporal process within a complex dynamical system. This body of work has generally been concerned with modeling the evolution of terrorist organizations, understanding the escalation of violence, or quantifying the lethality or number of attacks within a geographical region and/or over a period of time [12–18]. In one of these studies, Python et al. focused on modeling the continuous phenomena (e.g., lethality and frequency) exhibited by terrorism in the Middle East [19]. Li et al. similarly considered terrorism as "an unstable system of interdependent events" and treated it as a type of crime [20]. Prior crime prediction models have had some success when applied to urban, high-crime areas like

Chicago and New York City [21, 22], but these solutions do not necessarily translate to predicting terrorism, which is much more narrowly defined [2]. Although Li et al. applied their crime prediction model to terrorism the Middle East, neither theirs nor the other aforementioned works have considered terrorism in the United States, where attacks are much more sparse and the dynamical processes consequently more difficult to observe. For example, according to the GTD Iraq experienced 2,466 terrorist attacks in 2017, 1,154 of which were perpetrated by a single organization: the Islamic State of Iraq and the Levant (ISIL). However, in the same year, the United States experienced just 65 attacks, only five of which were perpetrated by a known organization [2]. Models that rely on systematic patterns exhibited by organizations or interdependence between events are unlikely to succeed in such a context.

Another vein of related work is the use of news data to predict social unrest. Qiao et al. used mentions of protests encoded in the Global Database of Language, Events, and Tone (GDELT) [23] to predict social unrest events in Southeast Asia [24]. Galla et al. extracted a set of longitudinal features from GDELT, which they used to predict social unrest events for a given location on a given date [25]. GDELT has also been recently utilized in other related applications, including modeling global peace [26].

In context of both the advances and limitations of these prior studies, we present a machine learning framework for predicting terrorist attacks in the United States using the GTD in conjunction with news data from GDELT. Our **key contributions** include the following:

1. We are, to the best of our knowledge, the first to use machine learning to predict the occurrence of individual terrorist attacks in the United States. We demonstrate that by framing terrorism prediction as a binary classification problem within discrete spatiotemporal units (i.e., did an attack take place on day *X* in location *Y*?) rather than a continuous process (i.e., the evolution of terrorist organizations over time, or the growth of terrorist activity in a region), we can model terrorism in a context where attacks are much less frequent and systematic than other regions that have previously been the focus of predictive tasks.

2. We show that large-scale news data can be successfully utilized in terrorism modeling. This is particularly noteworthy in a context where historical terrorism data is extremely sparse, and suggests many avenues for future work, such as incorporating news data into other difficult machine learning problems and studying the relationship between news trends and terrorist activity.

3. We present a simple but novel feature representation method, K-S Moving Average, which uses a Kolmogorov-Smirnov test [31] to choose an optimal observation window for each feature. The best model (a Random Forest using the K-S Moving Average) AUROC scores $\geq 0.667$ on four of five states ($p \leq .0001$ w.r.t. random guessing), outperforming other algorithms and representation methods—including several neural networks—on the classification problem.

4. We thoroughly evaluate a set of baselines and proposed models on the five states impacted most by terrorism between 2015 and 2018—New York, California, Texas, Florida, and Washington—and discuss the implications of our results for the future application of machine learning to the study of terrorism.

## Materials and methods

In this section, we first formulate the problem of predicting a terrorist attack at a given location and time. We then describe our methods for data collection and preprocessing. Finally, we

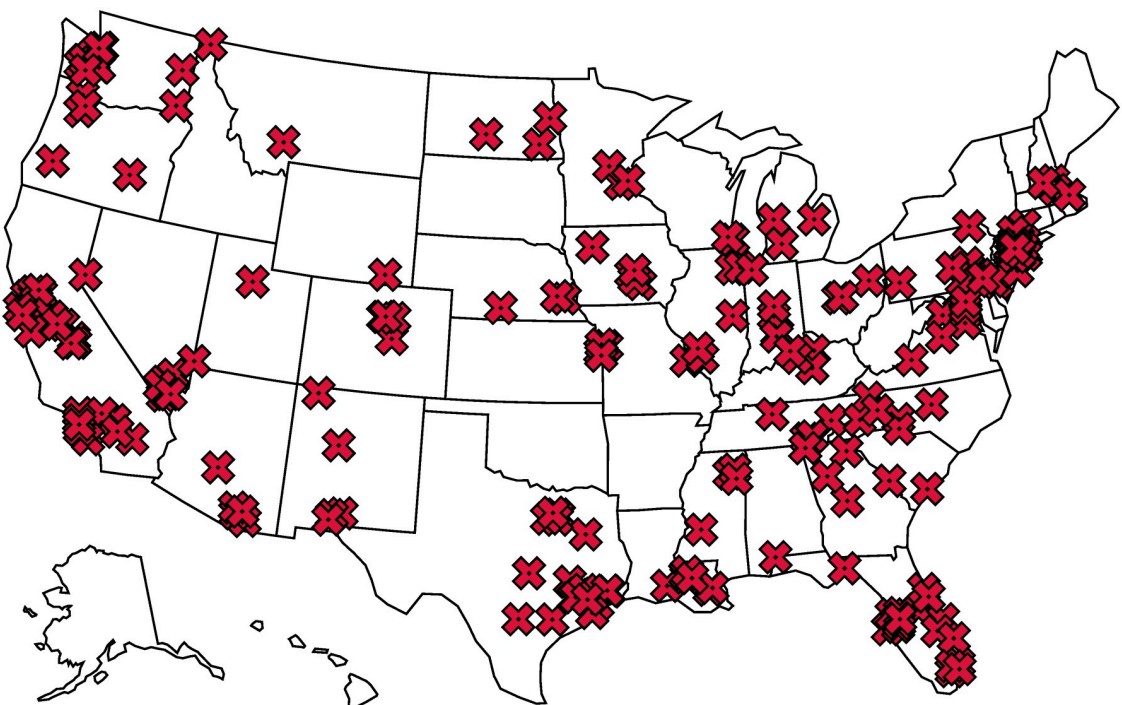

**Fig 2. Overview of feature extraction and representation.** Textual themes and CAMEO events from the Global Knowledge Graph and Events Database, respectively, were used to generate daily observations about each state. The ensemble methods learned from a moving average representation ($\Delta t$ and $\Delta t^*$) of the daily observations, and the neural network methods learned from a stacked representation. A red X indiciates an event (terrorist attack) while a white circle indicates a non-event (absence of terrorist attack), and each color represents a unique news feature.

introduce the relevant learning algorithms and approaches to feature representation. Fig 2 outlines our methodology.

## Problem formulation and notation

We formulate the task as a classification problem such that our goal is to learn a function $f: x \rightarrow y$, where $x$ is a set of input features and the binary output $y$ represents the occurrence or non-occurrence of a terrorist attack at a specific and on a given date. We consider as locations the set of 51 states in the United States (including Washington D.C., excluding Puerto Rico and other non-state territories), each with $m = 862$ features observed over $n = 1,413$ calendar days from February 18, 2015 through December 31, 2018, inclusive (as limited by data availability, please see below for more details). Let $X \in \mathbb{R}^{n \times m}$ be a feature matrix and $\vec{y} = \{0, 1\}^n$ be a label vector, such that $\vec{x}_i$ and $y_i$ denote the feature vector and label, respectively, that describe the $i^{th}$ day at the location of concern. Further, we use $X_{i,j}$ to represent the $i^{th}$ observation of the $j^{th}$ feature, $X_{i:k}$ to represent the $i^{th}$ through $k^{th}$ (inclusive) observations of all $m$ features, and $X_{*,j}$ to represent the column vector that contains all observations of the $j^{th}$ feature. Unless stated otherwise, we treat locations as independent such that each has a distinct feature matrix $X$ and label vector $\vec{y}$. Finally, we use square brackets [] to denote vectors.

## Data collection

**Feature extraction.** The Global Database of Events, Language, and Tone (GDELT) monitors worldwide print, broadcast, and online news in over 100 languages [23, 27]. It contains

two primary sources of information: the Global Knowledge Graph (GKG) and the Event Database. The GKG comprises over 10 TB and grows at a rate of over 500,000 records every day. Each record represents publishing information, mentions of persons and locations, textual themes, and other metadata extracted from each published article, video, or other item of news. The Event Database utilizes the CAMEO event coding framework to identify and categorize events from news in the GKG. A CAMEO event consists of an actor—an individual, country, identity group, or some other sociopolitical entity—performing an action on another actor [28]. Each action is classified under a taxonomy that includes categories such as "acknowledge or claim responsibility" (code 015) and "reject request for military aid" (code 1122). The Events Database records each event along with other extracted information, including the geographical location of the event and/or actors, political and religious affiliations for each actor, an estimate of the event's political significance, and the average sentiment of news items associated with the event. Taken together, the GKG and Events Database represent GDELT's effort to construct a global representation of "what's happening and how the world is feeling about it" [27]. We utilized version 2 of the GKG and Events Database, which were both released on February 18, 2015.

We utilized data from both the GKG and Events Database to construct the feature matrix for each location, in a manner similar to Galla et al. [25]. From the GKG, we considered 283 textual themes that are described in the GKG Category List [29]. While other themes have been added to the GKG, the aforementioned are the best-documented have been clearly recorded since the release of GDELT version 2. For each theme, we extracted all the news records associated with that theme that also mention the given location. To compute the first set of 283 features, which we call **theme counts**, we simply counted the relevant records in the GKG. For the next 283 features, which we call **theme sentiments**, we computed the average sentiment score of these records for each date. From the Events Database, we grouped all records by the 148 "base codes" at the second level of the CAMEO taxonomy [28]. The next set of 148 features, which we call **CAMEO counts**, is the count of news items associated with each event code for the given location, aggregated by publishing date. Our final set of 148 features, which we call **CAMEO sentiments** is the average sentiment score for all articles associated with each of the CAMEO codes, also aggregated by publishing date. In total, we considered 862 features, which collectively represent a location's sociopolitical climate on a given day.

**Label extraction.** The Global Terrorism Database (GTD), created and maintained by the University of Maryland, is an open-source database containing information on over 190,000 terrorist events around the world from 1970 through 2018 [2]. The GTD defines a terrorist attack as "the threatened or actual use of illegal force and violence by a non-state actor to attain a political, economic, religious, or social goal through fear, coercion, or intimidation" [30]. According to its codebook, an event must meet the following requirements to be classified as a terrorist attack and recorded in the GTD:

1. All three of the following criteria must be met:

    1. The incident must be intentional.

    2. The incident must entail some level of violence or immediate threat of violence.

    3. The perpetrators of the incident must be sub-national actors.

2. Two of the following three criteria must be met:

    1. The act must be aimed at attaining a political, economic, religious, or social goal.

2. There must be evidence of an intention to coerce, intimidate, or convey some other message to a larger audience (or audiences) than the immediate victims.

3. The action must be outside the context of legitimate warfare activities.

We considered all terrorist events that occurred in the United States on or after February 18, 2015—the release date of GDELT version 2. Of these 229 events, shown in Fig 1, 83.8% were successful, meaning that the attack produced some kind of tangible effect (such as an explosion), even if there were no casualties or significant consequences. 19.1% of the attacks resulted in at least one fatality, and in 27.2% at least one person was wounded. If multiple attacks were carried out in a continuous period of time or location, the GTD records them as a single incident. However, if either the time or location was discontinuous (even if the attacks were carried out in close proximity or by the same perpetrators) the GTD records them as multiple incidents. After accounting for this overlap in the 229 events, we extracted 208 unique location and date pairs as positive labels ($y_i = 1$) for the classification problem. We treated all location and date pair not recorded in the GTD as negative labels ($y_i = 0$).

## Observation windows

Recall that our goal is to predict future terrorist attacks. However, the features and labels described above represent news activity and the occurrence of an attack, respectively, for the same day. In a real-world scenario, the news features for a given day would not be available until after the attack occurred. Additionally, the features only consider news activity for a single day; this precludes the discovery of any long-range temporal patterns. In other words, in their raw form the GDELT features can only be used same-day terrorism detection, but not prediction.

To address both of these shortcomings, we performed an additional preprocessing step to generate new feature representations for each date and location pair using an **observation window** [25]. This window, represented by the hyperparameter $\Delta t \geq 1$, is the number of prior days for which news features are observed when predicting an attack. As outlined in Fig 2, we evaluated three distinct types of observation windows: a fixed-length moving average, a variable-length moving average, and a stacked representation. Each approach is detailed below.

**Fixed-length moving average.** In the simplest case, for each feature $X_{*j}$, we considered the unweighted mean of the previous $\Delta t$ days, i.e.

$$MA(X_{*j}, \Delta t) = \left[ \frac{\sum_{k=1}^{\Delta t} X_{i-k,j}}{\Delta t} : i \leq n \right]. \tag{1}$$

Note that when $\Delta t = 1$, the representation for a given location and date is simply the news features as observed from the previous day. Increasing $\Delta t$ allows us to observe a greater number of prior days and has the additional benefit of smoothing noise. However, it also blurs the distinctions between days and causes each feature to no longer be independently distributed; we address this problem in our discussion of Experimental Setup.

**Variable-length (K-S) moving average.** A significant limitation of the fixed-length approach is that it assumes a static value of $\Delta t$ for all features. We addressed this by implementing a more flexible version of the observation window, which chooses a preferred representation for each feature according to a two-sample Kolmogorov-Smirnov (K-S) test [31]. The K-S test computes the distance between the empirical distribution function (EDF) of two samples in order to test the null hypothesis that the two samples are drawn from the same distribution. Intuitively, we use the K-S test to choose a representation for each feature that maximizes the difference between the EDF for both classes for that feature. The full procedure is detailed in

Algorithm 1. In summary, we first specify a maximum observation window $\Delta t^*$. Next, for each feature we compute its moving moving average (Eq 1) for each $t \leq \Delta t^*$. Then for each feature and value of $t$ we perform a K-S test, where the first sample is the set of observations for that feature for non-events ($y_i = 0$) and the second sample is the set of observations for events ($y_i = 1$). Finally, we choose the representation for each feature (i.e., value of $t$) that maximizes the distance between the empirical distribution functions of events and nonevents (i.e., returns the minimum $p$-value).

**Algorithm 1** K-S Moving Average

```
Require: A feature matrix X ∈ ℝⁿˣᵐ, a label vector ȳ = {0,1}ⁿ, a maximum
  observation window Δt* ≥ 1
Ensure: A processed feature matrix X
 1: for j ← 1 to m do
 2:   p_min ← 1.0      ▷ Store the minimum p-value
 3:   a ← X_*,j        ▷ Store the original observations for feature j
 4:   for t ← 1 to Δt* do
 5:     a' ← MA(a, t)      ▷ Compute the current moving average via Eq 1
 6:     p ← KS({a'_i : y_i = 0}, {a'_i : y_i = 1})      ▷ Compute the current p-value
 7:     if p < p_min then
 8:       p_min ← p      ▷ Store the new minimum
 9:       X_*,j ← a'      ▷ Replace the values for feature j
10: return X
```

This procedure requires computing a moving average over $n$ instances, which takes $O(n)$ time, exactly $\Delta t^*$ times for each of the $m$ features. Therefore, the time complexity of Algorithm 1 is $O(\Delta t^* nm)$. Since the window is computed independently for each feature, the algorithm can be easily parallelized. Fig 12 in S1 Appendix reports empirical runtimes on the data used in this study.

**Stacked representation.** Both moving average approaches described previously are unweighted and thus assume each of the values within the observation window are equally important. While this may be suitable for simpler learning algorithms, neural networks are well-equipped to discover more complex patterns by generating their own (hidden) representations. Therefore, when serving input to neural networks we do not precompute representations using either of the moving average approaches described above; instead, we represent each instance as stacked feature vectors from the previous $\Delta t$ days, i.e. $X_{i-\Delta t:i-1}$ for some arbitrary day $i$. By learning a mapping from these stacked inputs to a hidden representation, the neural network is essentially learning its own observation window. We discuss this in more detail in the following section.

## Learning algorithms

Recall that our task is to learn a function $f: x \rightarrow \{0, 1\}$ for some input $x$. Toward this end we explored two families of machine learning algorithms, ensembles and neural networks, which we detail below.

**Ensembles.** An ensemble is combination of multiple classifiers. When individual classifiers are independent and diverse, the ensemble can mitigate the weaknesses and limitations of a single classifier. This characteristic has enabled ensembles to provide state-of-the-art performance on a number of machine learning tasks [32]. We evaluated the following three ensembles:

1. XGBoost: a tree-based ensemble that introduced a number of innovations to gradient boosting algorithms. XGBoost has achieved state-of-the-art performance on a number of difficult problems [33].

2. Random Forest: a collection of unpruned decision trees, each grown from a random and uniformly sampled subspace (of features), that uses a weighted voting mechanism to classify each instance [34]. The Random Forest has found success in a range of machine learning tasks and established itself as a strong general-purpose ensemble [32].

3. AdaBoost: a collection of decision stumps that are each trained on a single bootstrap (a small sample drawn with replacement from the training set). AdaBoost's key innovation is that it iteratively updates the sampling distribution, increasing the probability of selection in the next bootstrap for instances that were misclassified by the previous decision stump [35]. This procedure intuitively pushes the ensemble to "focus" on more difficult instances; however, it also makes AdaBoost very sensitive to noise.

**Neural networks.**    A neural network propagates an instance through layers of nodes, or neurons, to compute a label. Training a neural network is the process of learning the set of edge weights that minimizes error on the training data with respect to some loss function. The power of neural networks lies in their use of various configurations of hidden (intermediate) layers to generate new representations of the input. As such, neural networks have become state-of-the-art in many complex learning tasks—especially those that involve learning nonlinear functions [32]. A neural network is a function that classifies an input $x$ according to the following:

$$f(x) = \sigma(Wh(x) + \vec{b}),\tag{2}$$

where $W$ is a real-valued weight matrix and $\vec{b}$ is a real-valued bias vector that are both learned via backpropagation; $\sigma$ is the sigmoid activation function, and $h(x)$ is the learned hidden representation of the raw input $x$. In this context, what distinguishes classes of neural networks is how they compute $h(x)$. For brevity and in this section only, we use $X_{i-\Delta t:i-1}$ to represent the matrix form of the stacked representation for an arbitrary day $i$ and observation window $\Delta t \geq 1$. Given this definition of $X'$ we utilized the following neural network functions:

1. Feedforward Neural Network (FFNN): a basic architecture in which inputs are propagated strictly forward from the input layer, through a series of $L$ hidden layer(s), and finally to the output layer. We utilize two distinct FFNN architectures:

    1. One hidden layer ($L = 1$, not pictured): computes $h(X')$ via a single dense layer, i.e.

    $$h(X') = ReLU(W_0[X'_{*,1}, ..., X'_{*,m}] + \vec{b_0}),\tag{3}$$

    where $ReLU$ is the rectified linear unit activation function, $W_0 \in \mathbb{R}^{m \times k}$ is a learned weight matrix with $k$ (a hyperparameter) neurons in the hidden layer, $[X'_{*,1}, ..., X'_{*,m}]$ represents the vector concatenation of all observations of all features in $X'$, and $\vec{b_0} \in \mathbb{R}^k$ is a learned bias vector for the hidden layer.

    2. Two hidden layers ($L = 2$, pictured in Fig 2): computes a hidden representation for each feature independently, then uses each feature's new representation to compute a joint

hidden representation via a dense layer:

$$h(X') = ReLU(W_0[g(X'_{*,1}), ..., g(X'_{*,m})] + \vec{b_0}), \tag{4}$$

$$g(X'_{*,j}) = ReLU(W_j X'_{*,j} + \vec{b_j}). \tag{5}$$

Here $W_j$ is the weight matrix and $b_j$ the bias vector for the $j^{th}$ feature.

2. Long Short-term Memory (LSTM): an architecture in which inputs are propagated through a series of recurrent layers in order to learn sequential dependencies [36]. LSTMs have found success in time series classification, machine translation, and a number of other tasks in which the input can be modeled as a sequence. We use a standard sequential LSTM to compute a hidden representation for each day $t < \Delta t$ according to:

$$h_t(X') = tanh(W_t[\vec{x_t}, h_{t-1}(X'_{1:t-1})] + \vec{b_t}), \tag{6}$$

where $tanh$ is the standard $tanh$ activation function and $\vec{x_t}$ is the feature vector for day $t$ in $X'$.

We experimentally evaluated several other state-of-the-art neural network architectures, including models based on attention [37] and convolutions, which are often used in time series classification and have the benefit of being translation invariant [38]. However, we found that these additional architectures performed empirically worse than both the FFNN and LSTM, so we do not consider them in the rest of this study.

## Addressing class imbalance

When trained on imbalanced data, the predictions of machine learning models can be biased toward the majority class. We utilized the following techniques to address this problem:

**SMOTE.**  Synthetic minority oversampling technique (SMOTE) generates artificial examples of the minority class in order to balance the training set [39–41]. We utilized SMOTE in all the ensembles by oversampling the minority class (events) such that both classes had equal representation.

**Weighted cross-entropy.**  Because backpropagation trains a neural network by computing the gradient with respect to a loss function, a useful technique in imbalanced settings is to weigh more heavily the loss contributions from the minority class. Toward this end we utilized a weighted version of cross-entropy (log loss). Consider the standard definition of binary cross-entropy, i.e.

$$H(\vec{y}) = -\frac{1}{n}\sum_{i=0}^{n}(y_i log_2(p(y_i)) + (1 - y_i)log_2(1 - p(y_i))), \tag{7}$$

where $p(y_i)$ is the probability output by the model that example $y_i = 1$ (is an event). We added a term $\alpha = \frac{\sum_{i=0}^{n} y_i}{n}$ to weigh the contribution of each training example by the inverse of its class representation:

$$H_w(\vec{y}) = -\frac{1}{n}\sum_{i=0}^{n}((1 - \alpha)y_i log_2(p(y_i)) + \alpha(1 - y_i)log_2(1 - p(y_i))). \tag{8}$$

This weighting allows for the set of events and the set of nonevents to each contribute half to the potential loss.

We explored a number of other options for addressing class imbalance, including under- and over-sampling, sample and class weighting, and state-of-the-art loss functions for imbalanced classification [42, 43]. However, we found that SMOTE and weighted cross-entropy provided the best performance in almost all cases for the ensembles and neural networks, respectively, so all results presented below use these techniques. Fig 11 in S1 Appendix summarizes the performance of other strategies.

## Experimental setup

We first evaluated each model on the states that had experienced the greatest number of attacks: New York, California, Texas, Florida, and Washington (Table 1). While we also evaluated additional locations, our results demonstrate that in these cases the attacks were too sparse to learn effective models. Therefore, we dedicate most of our analysis to these five states.

As baselines, we evaluated a Random Guesser (assigned a random probability to each testing example), a single Decision Tree, a support vector machine (SVM) using an RBF kernel, and a Logistic Regression classifier. For all tree-based methods we used Gini index as the splitting criterion, and for each ensemble we used 3,000 estimators. We trained all neural networks for 100 epochs using stochastic gradient descent with a learning rate of $1e^{-4}$, a decay parameter of $1e^{-6}$, a batch size of 32, and Nesterov momentum. All hidden layers (except the first hidden layer in the two-layer FFNN architecture) contained 8,000 neurons, and each hidden layer in the LSTM model contained 1,024 neurons. For the neural network models we used min-max normalization to scale the input. Beyond computing the observation windows, no further preprocessing was required for the ensembles. We tested the fixed-length moving average approach with the ensemble methods and the stacked representation with the neural networks. However, due to the success of the K-S moving average ($\Delta t^*$) with the ensembles, we also evaluated this representational method with the neural networks.

We evaluated each model using 5-fold cross-validation. This was straightforward for the neural network models and when $\Delta t = 1$, since each observation was independent. However, when using either moving average approach we considered the following in order to maintain proper separation between the training and testing sets:

1. When testing the variable-length moving average ($\Delta t^*$), we used only the training data to compute the optimal observation windows for each feature. We then computed moving averages on the testing set using the same window lengths that were learned from the training set.

2. When testing either type of moving average, we removed any examples from the training set whose values were dependent on examples in the testing set. For example, consider the case where $\Delta t = 14$ and the testing fold is the 282 days from Mar. 25 through Dec. 31, 2018. The feature values for Mar. 25 will be highly correlated with the values for Mar. 11–24, which are in the training set, since their moving averages are computed using at least one of the same observations. We resolved this by dropping Mar. 11–24 from the training set. This approach handicapped the moving average models (since they are provided with fewer training examples), but ensured that the testing sets are consistent across all models.

Finally, we measured model performance on each testing fold using AUROC. An ROC curve plots a model's true positive rate against its false positive rate at various discrimination thresholds, and is thus a useful measure of classification performance in an imbalanced setting. An AUROC of 1.0 indicates a perfect classifier, while a random guesser converges to 0.5. We repeated each 5-fold cross-validation experiment 10 times and report both the mean and standard deviation AUROC values for each model. To compare the performance of individual

**Table 2. Summary of models whose outputs were statistically significant on at least 3 states.**

| Model | Observation Window | $p < .01$? | | | | | AUROC Summary | Proposed Rank |
|---|---|---|---|---|---|---|---|---|
| | | NY | CA | TX | FL | WA | | |
| Random Forest | $\Delta t^* = 14$ | ✓ | | ✓ | ✓ | ✓ | .636 ± .167 | 1 |
| XGBoost | $\Delta t^* = 14$ | ✓ | | | ✓ | ✓ | .627 ± .161 | 2 |
| FFNN | $\Delta t = 7, L = 1$ | | ✓ | ✓ | ✓ | | .597 ± .178 | 3 |
| Random Forest | $\Delta t = 1$ | ✓ | | ✓ | ✓ | | .582 ± .183 | 4 |

AUROC Summary is the mean and standard deviation of the AUROC values reported in Table 3 across all states. $p$-values were computed w.r.t. Random Guesser using a one-tailed Welch's unequal variances t-test (see Table in S1 Table for more details).

classifiers, we performed a one-tailed Welch's unequal variances test [44] on the outputs for all testing folds using 0.01 as our significance threshold. All experiments utilized Python 3.7.3 in conjunction with scikit-learn 1.0.2 (ensembles), Keras 2.2.4 with TensorFlow 2.0 (neural networks), and SciPy 1.7.3 (statistical analysis).

## Results and discussion

Table 2 summarizes the performance of the models that produced statistically significant predictions on at least three of the five states that experienced the most attacks. Table 3 shows the classification results for observation windows of 1 and 14 days for the baselines and ensembles, and 1 and 7 days for the neural networks. While some other observation windows produced better results for individual states, and as we discuss below, many models were very sensitive to the value of this parameter. Those shown in Table 3 produced at least one model that was strong across several states, and thus seemed the most useful and representative for evaluating how well different types of algorithms can perform on this problem. The standard deviation for each complete 5-fold cross-validation experiment was small ($< 0.025$ in all cases), but the inter-fold standard deviation was high in many cases. For example, on New York, the Random Forest ($\Delta t^* = 14$) model consistently performed well on the first testing fold (.726 ± .036) but struggled on the fifth fold (.615 ± .036). This is not surprising, given that individual events are difficult to predict and that the small number of events in our data set means it is less likely that each training and testing fold will reflect the true distribution. In spite of this, we found that multiple models achieved results that are noteworthy and significantly better than random.

The Random Forest with $\Delta t^* = 14$ achieved an AUROC $\geq .667$ for every state except California, on which all the ensembles performed poorly. The neural networks performed better on California, but this was exchanged for poor performance on New York. The neural network that performed best on average was the FFNN with $\Delta t = 7, L = 1$, which achieved AUROC $\geq .650$ on three of five states. The other neural networks performed well on two states—California and Washington for the FFNN with $\Delta t^* = 7, L = 2$, and Texas and Florida for the LSTM—but poorly on the others. Except for the SVM on California, none of the baselines performed well. Overall, we consider the Random Forest ($\Delta t^*$) to be the strongest model, followed closely by XGBoost ($\Delta t^*$), which produced very similar results on four of five states. The greatest difference was on Texas; however, this difference in AUROC may not be significant ($p = 0.069$), so we consider both models to be roughly equivalent in terms of overall performance. Of the neural networks, the FFNN ($L = 1$) produced statistically significant results on the greatest number of states (3). Since they represent the strongest overall performance of each family of

**Table 3. Summary of classification results.**

| | Model | Observation Window | AUROC (5-fold cross-validation) | | | | |
|---|---|---|---|---|---|---|---|
| | | | NY | CA | TX | FL | WA |
| Baselines | Random Guesser | — | .500 ± .121 | .499 ± .149 | .501 ± .163 | .499 ± .155 | .502 ± .203 |
| | Decision Tree | $\Delta t = 1$ | .506 ± .037 | .481 ± .024 | .497 ± .030 | .495 ± .028 | .513 ± .055 |
| | | $\Delta t = 14$ | .493 ± .056 | .496 ± .084 | .504 ± .060 | .480 ± .018 | .508 ± .090 |
| | | $\Delta t^* = 14$ | .523 ± .062 | .490 ± .050 | .513 ± .043 | .487 ± .015 | .497 ± .030 |
| | SVM | $\Delta t = 1$ | .423 ± .196 | **.714 ± .071** | .444 ± .082 | .472 ± .080 | .499 ± .241 |
| | | $\Delta t = 14$ | .558 ± .196 | .473 ± .243 | .483 ± .195 | .505 ± .309 | .497 ± .234 |
| | | $\Delta t^* = 14$ | .453 ± .092 | **.644 ± .218** | .527 ± .190 | .411 ± .147 | .464 ± .230 |
| | Logistic Regression | $\Delta t = 1$ | .439 ± .112 | **.589 ± .118** | .462 ± .134 | .436 ± .077 | .429 ± .233 |
| | | $\Delta t = 14$ | .534 ± .220 | .504 ± .083 | .529 ± .161 | .426 ± .103 | .439 ± .308 |
| | | $\Delta t^* = 14$ | .454 ± .114 | **.586 ± .111** | .450 ± .176 | .496 ± .171 | .352 ± .158 |
| Ensembles | AdaBoost | $\Delta t = 1$ | **.623 ± .079** | .436 ± .110 | **.574 ± .117** | .459 ± .093 | .376 ± .122 |
| | | $\Delta t = 14$ | .500 ± .243 | .360 ± .189 | .401 ± .136 | .421 ± .126 | .538 ± .253 |
| | | $\Delta t^* = 14$ | **.668 ± .073** | .463 ± .135 | .589 ± .136 | .472 ± .123 | .544 ± .171 |
| | Random Forest | $\Delta t = 1$ | **.604 ± .118** | .504 ± .065 | **.721 ± .184** | .591 ± .140 | .500 ± .248 |
| | | $\Delta t = 14$ | **.631 ± .093** | .390 ± .133 | .530 ± .166 | **.623 ± .120** | .591 ± .261 |
| | | $\Delta t^* = 14$ | **.685 ± .057** | .466 ± .060 | **.682 ± .196** | **.685 ± .101** | **.667 ± .214** |
| | XGBoost | $\Delta t = 1$ | **.589 ± .070** | .479 ± .123 | **.692 ± .212** | .537 ± .134 | .570 ± .214 |
| | | $\Delta t = 14$ | **.663 ± .117** | .439 ± .168 | .387 ± .155 | **.601 ± .117** | .490 ± .227 |
| | | $\Delta t^* = 14$ | **.703 ± .084** | .477 ± .120 | .600 ± .202 | **.675 ± .076** | **.679 ± .173** |
| Neural Nets | FFNN | $\Delta t = 1, L = 1$ | .393 ± .065 | .584 ± .100 | .454 ± .199 | .420 ± .099 | .409 ± .149 |
| | | $\Delta t = 7, L = 1$ | .414 ± .110 | **.672 ± .095** | **.650 ± .118** | **.617 ± .098** | .605 ± .252 |
| | | $\Delta t = 7, L = 2$ | .428 ± .095 | **.680 ± .073** | .528 ± .127 | .362 ± .144 | **.648 ± .201** |
| | | $\Delta t^* = 7, L = 1$ | .479 ± .219 | **.568 ± .077** | **.591 ± .175** | **.552 ± .039** | .546 ± .229 |
| | LSTM | $\Delta t = 7$ | .374 ± .078 | .582 ± .153 | **.664 ± .212** | **.574 ± .168** | .563 ± .270 |

Reported values are the means and standard deviations of the cross-validation experiments. **Bold values** indicate results that are statistically significant ($p < .01$ w.r.t. Random Guesser using a one-tailed Welch's unequal variances t-test; see Table in S1 Table for more details), and underlined values indicate the best-performing model in each group of classifiers for each state.

models, we focus the remainder of our discussion on the performance of the Random Forest and FFNN ($L = 1$) and refer to them as *RF* and *FFNN*, respectively.

## Observation windows

Fig 3 depicts model performance as a function of observation window (each tick on the x-axis is a separate model) for RF and FFNN. Ideally, a model should be robust to small changes in the observation window and benefit (or at least not suffer) from increasing its length. We note that when using the K-S Moving Average ($\Delta t^*$) representation, the trend of RF's performance is relatively stable with respect to increased window length. FFNN and the standard Moving Average ($\Delta t$), on the other hand, are both relatively sensitive to small changes to the observation window. This instability is likely attributable to noise, and suggests that RF with the K-S Moving Average is less prone to overfit. From this result, we additionally note that the trend and optimal window varies between states. While this suggests that a predictive model may need to be tuned separately for each state, it is also possible that some of this variation is due to noise and/or only having few events per state.

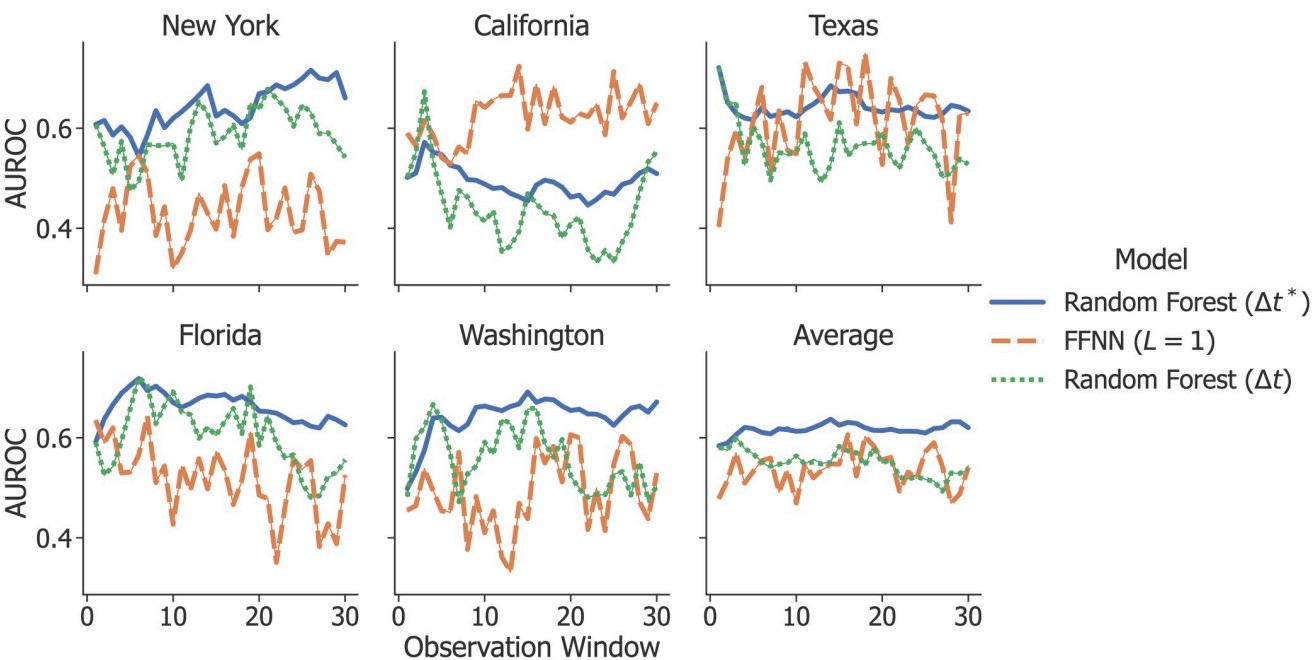

**Fig 3. Classification results for different observation windows.** These show that the optimal window varies between states. The x-axis represents the observation window (number of previous days considered in the feature representation for each instance), and the y-axis represents the average AUROC computed using 5-fold cross-validation.

## Temporal locality of attacks

Some attacks recorded in the GTD are clustered in time and space. For example, in 2016 attacks were carried out in New York on August 9 (anti-Semitic extremists detonated explosives outside the houses of two Rabbis in New York City), August 10 (an arsonist targeted a private residence in Endicott), and August 13 (a man shot and killed an Imam and his assistant outside a mosque in New York City). Additionally, there are a few attacks that spanned a number of days and/or targets, such as in October 2018, when pro-Trump extremists sent pipe bombs to five different targets on four different dates. To test whether temporal locality had biased the model outputs, we recorded for each attack the predicted probability ($P(y = 1)$) and the number of days that had elapsed since the previous attack in that state. Fig 4 visualizes this distribution for RF and FFNN along with a regression line for each state. These results, along with the correlation coefficient from Spearman's rank test ($r_s$), indicate that RF still performs well on many of the attacks that are not close together in time, which is an important characteristic of a robust model. FFNN performed more poorly on attacks that were farther apart in time, which likely contributed to its weaker overall performance.

## Number of attacks in training and testing

The fact that our data set is both small and highly imbalanced is almost certainly a limiting factor in model performance. Additionally, the temporal nature of our data limits our ability to use stratified cross-validation, so the events are not distributed evenly across training and testing folds. Fig 5 plots the average AUROC of each testing fold as a function of the number of events in the corresponding training fold for RF and FFNN, respectively. When computed across all states, the line of best fit and Spearman's rank coefficient ($r_s$) for these results

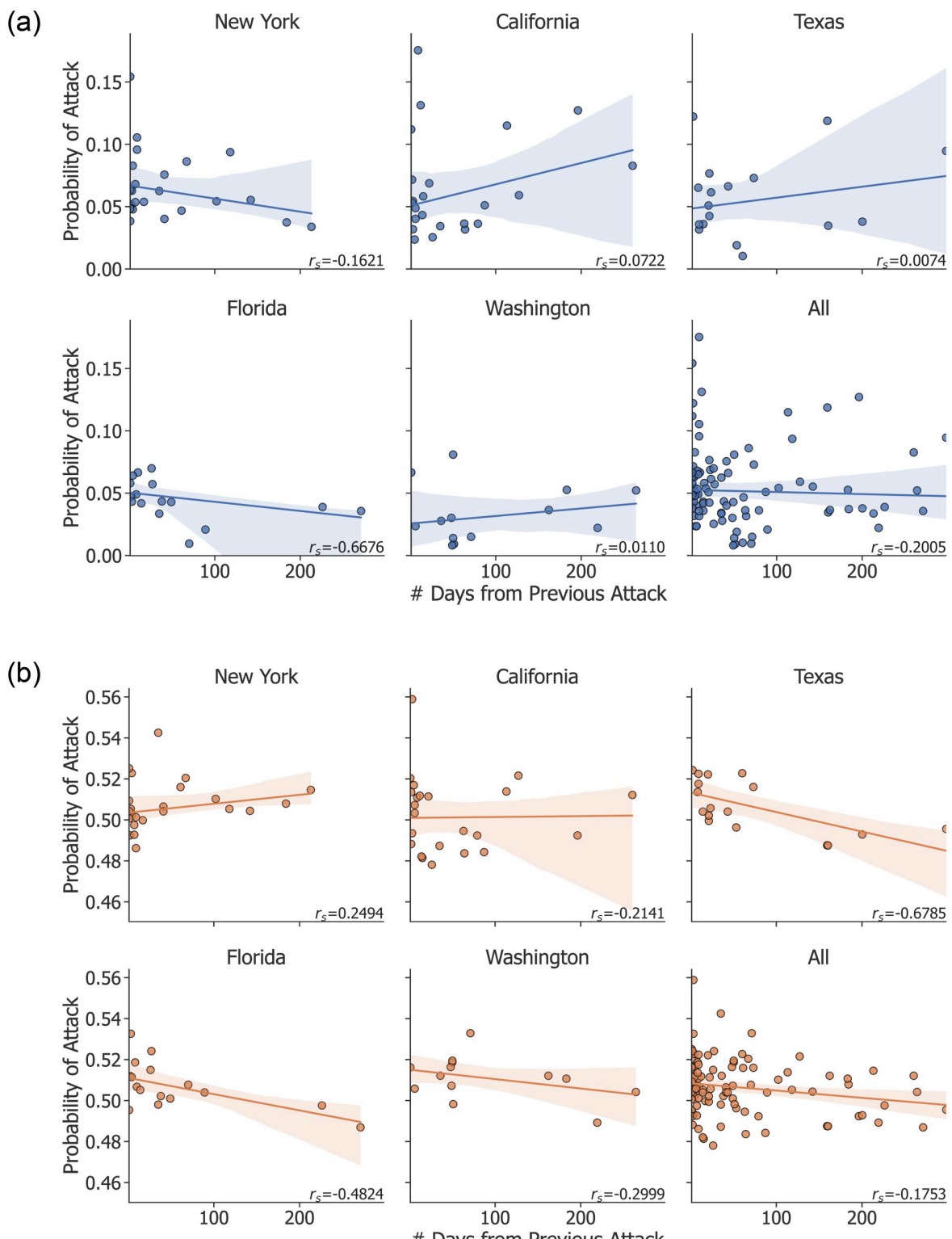

**Fig 4. Predicted probabilities for each attack.** The probabilities for RF (top, blue) and FFNN (bottom, yellow) are plotted as a function of distance in time from the previous attack in that state. The shaded region is the confidence interval for the regression line, and $r_s$ is Spearman's rank correlation coefficient. The difference in scale in the y-axis between RF and FFNN is caused by differences in the training process for each model.

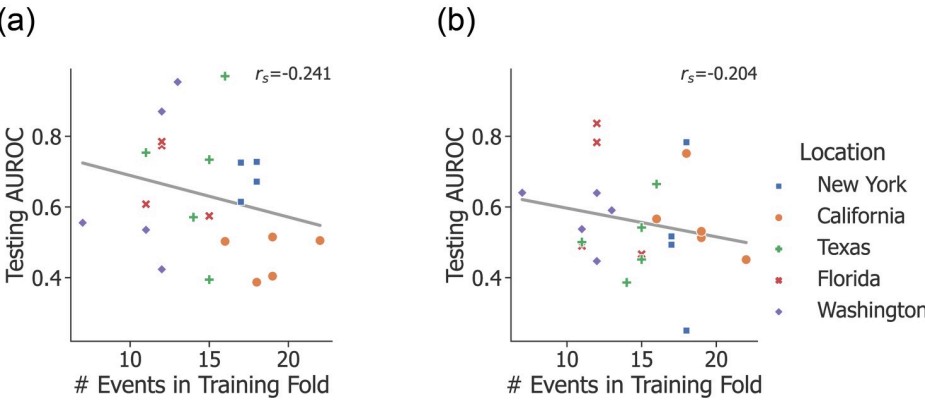

**Fig 5. AUROC of each test fold as a function of the number of attacks in the corresponding training fold.** These results for RF (left) and FFNN (right) show that having more attacks in the training set does not necessarily increase performance on the test set. The grey lines are regression lines, and $r_s$ is Spearman's rank correlation coefficient.

surprisingly suggest a negative correlation—i.e., that a greater number of events in the training and testing sets is correlated with worse performance. However, in the case of RF, we can see that the folds from California—on which RF's predictions were close to random guessing—contribute significantly to this result since they contain many events. Without California, $r_s$ increases from −0.241 to 0.136 for RF—the positive correlation we would expect. Continued efforts to collect both news and terrorism data will very likely drive further performance gains. However, for both RF and FFNN it seems that the lack of events or data is not the only limiting factor. It is likely that data quality and/or noise are additional barriers, ones that are more pronounced given the small number of events.

## Feature types

Measuring the predictive power of each group of features can help us understand model performance as well as identify areas for future improvements. Toward this end, we retrained RF and FFNN using different combinations of feature groups. Fig 6 shows the baseline classification results on each state along with results from four additional feature-filtered models, and is the basis for the following observations:

1. In general, the importance of each feature group varied between states. This affirms the need to consider a unique model for each state.

2. The only feature group that improved performance in all cases was CAMEO counts. Further, in most states the model without CAMEO counts performed the worst. This implies that CAMEO counts are, on average, the most potent feature group.

3. In some cases, dropping a feature group improved model performance; for example, RF's predictions on California. This is further evidence of the impact of noise on classification results, and creates an important point of investigation for any future work that would seek to optimize predictions for a particular state or location.

## Characteristics of attacks

Terrorist attacks vary widely with respect to responsible party, target, method and other properties. Some types of attacks may be easier or harder to predict, or even characterized by

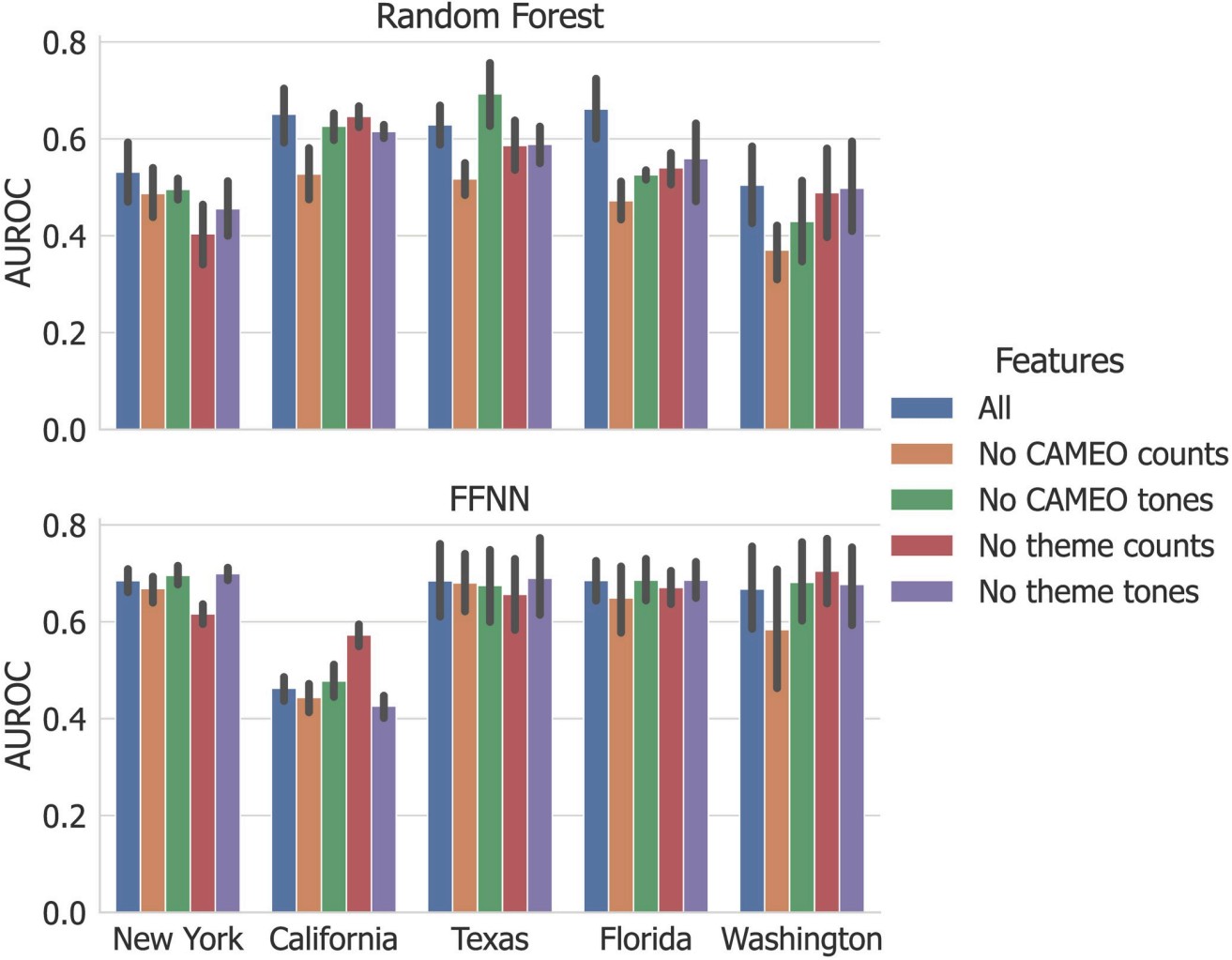

**Fig 6. Classification results for models trained on subsets of features.** For both RF (top) and FFNN (bottom), each state was evaluated using the baseline model along with four additional models, each of which was trained without one of the major feature groups. Error bars represent standard deviations between folds in each 5-fold cross-validation experiment.

different feature distributions. We used a Kruskal-Wallis H-test [45] to test whether the median predictions from RF or FFNN differed significantly by each of the GTD's prominent categorical features: attack type (e.g., armed assault, bombing), primary weapon type (e.g., firearms, explosives), target type (e.g., military, religious institutions), and responsible group. As Fig 7 shows, each test produced a $p$-value $>0.1$, so we cannot conclude that there is a correlation between the attack type and prediction success. Similar trends held for weapon type, target type, and responsible group.

## Prediction windows

Our baseline task is to predict the occurrence of a terrorist attack in a given state on a single day. Some prior work found success in making predictions over coarser periods of time, such as a week [25]. However, we found this was not the case for our problem. We evaluated the following three methods for performing predictions over a given prediction window $\Delta p > 1$:

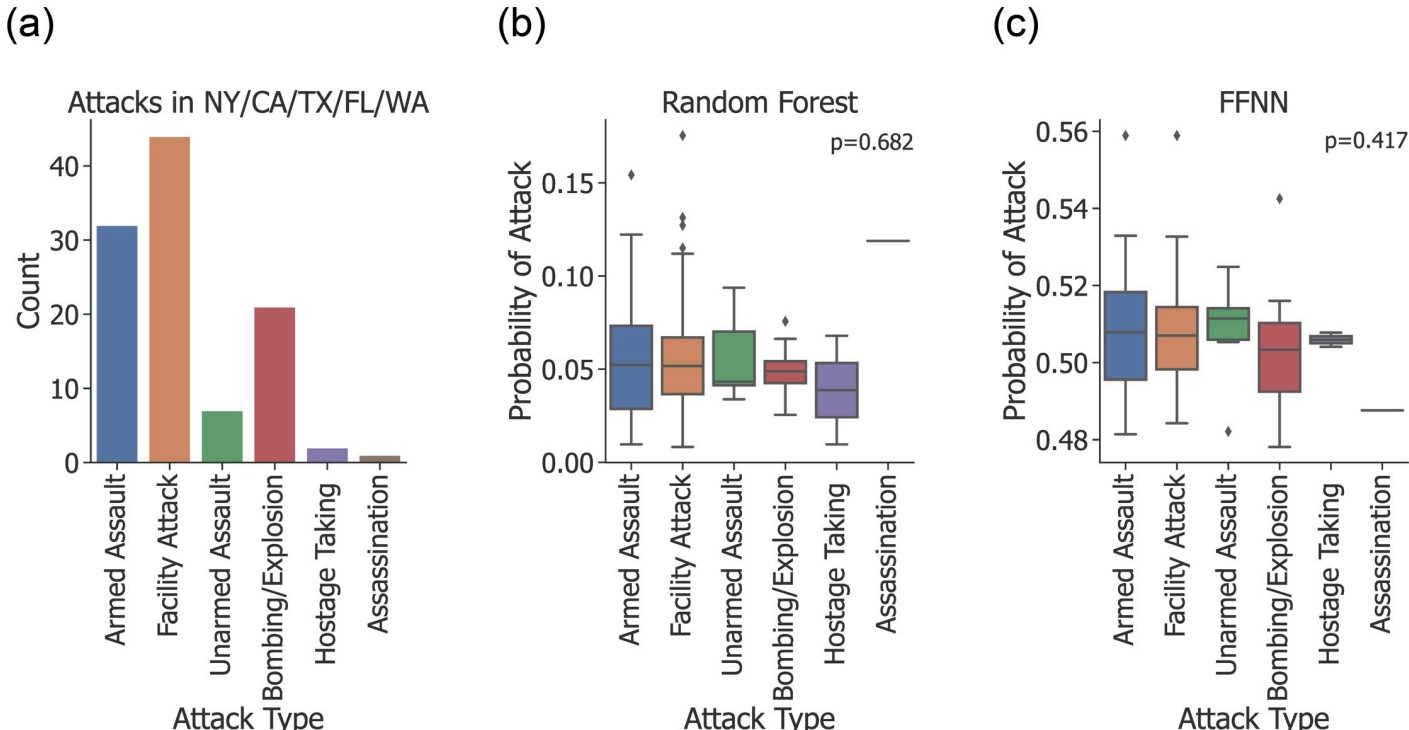

**Fig 7. Distribution of event count and predicted probabilities ($P(y = 1)$) by attack type.** The difference in scale in the y-axis between RF and FFNN is caused by differences in the training process for each model. In the box plots, each box represents observations within 1 standard deviation of the mean, which is represented by the horizontal line. Error bars represent observations within 3 standard deviations of the mean, while individual points represent outliers. (**a**) Distribution of attacks by type as recorded in the GTD. (**b**) Distribution of RF's predicted probabilities. (**c**) Distribution of FFNN's predicted probabilities.

1. Label propagation: we propagated event labels backward to the previous $\Delta p$ observations. For example, given an attack on Jan. 3 and $\Delta p = 3$, we labeled Jan. 1, 2, and 3 as attacks.

2. Date aggregation: we aggregated the news features from $\Delta p$ dates into a single observation. For example, given $\Delta p = 3$, we combined Jan. 1, 2, and 3 into a single observation: Jan. 1–3. We used mean pooling to generate the feature values for each aggregated observation.

As shown in Fig 8, these approaches resulted in worse performance, with only a few exceptions. AUROC measures the trade-off between true positives and false positives, so whatever improvements larger prediction windows might have provided in successfully predicting attacks is counterbalanced by an increase in false positives. This is not surprising, given that all of these methods run the risk of blurring the observed distinctions between classes. Date aggregation has the additional disadvantage of further reducing the number of training examples. A potential item of future work is to develop a more sophisticated method for making predictions over coarse periods of time without blurring the distinction between individual examples.

## Learning from multiple states

One of our key challenges is a small and imbalanced data set. While our results thus far have shown that each state should be modeled independently, we also hypothesized that it could be possible to transfer knowledge between models. To evaluate the feasibility of such an approach, we trained a new set of models by combining data from multiple states. For example, when

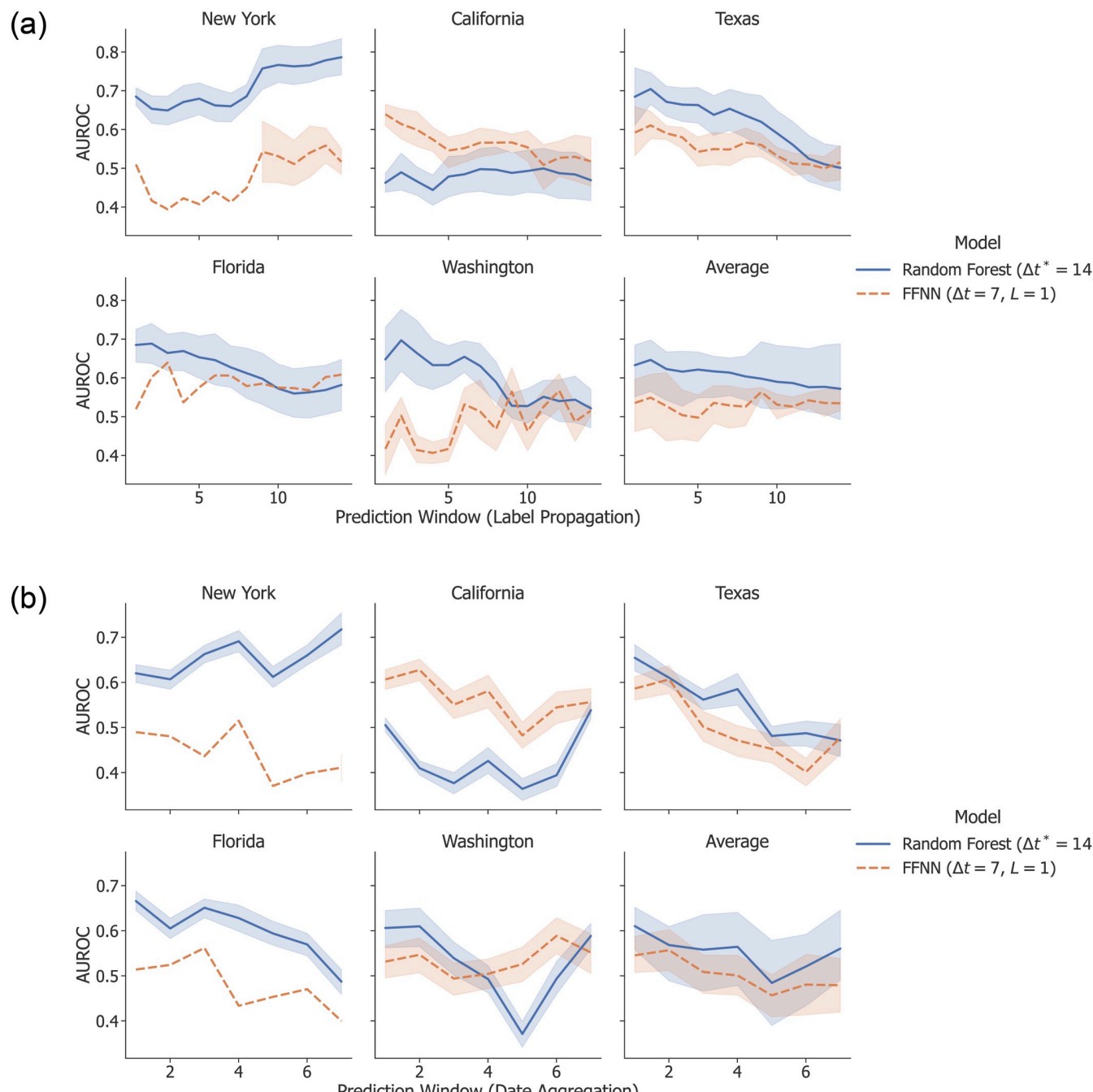

**Fig 8. Classification performance for models using larger prediction windows.** The performance of both label propagation (top) and date aggregation (bottom) suggests that using coarser date representations hinders the models' ability to discriminate between classes. In each plot, AUROC is shown as a function of Δp (the prediction window) for the given state. Shaded regions represent the standard deviation across testing folds.

testing the New York model at baseline, we only included the training examples from New York; however, in these experiments we also supplemented the training set with additional data from another state (e.g., California) before evaluating on the New York testing set. As the heatmap in Fig 9 shows, this only proved beneficial on average for California. This is not

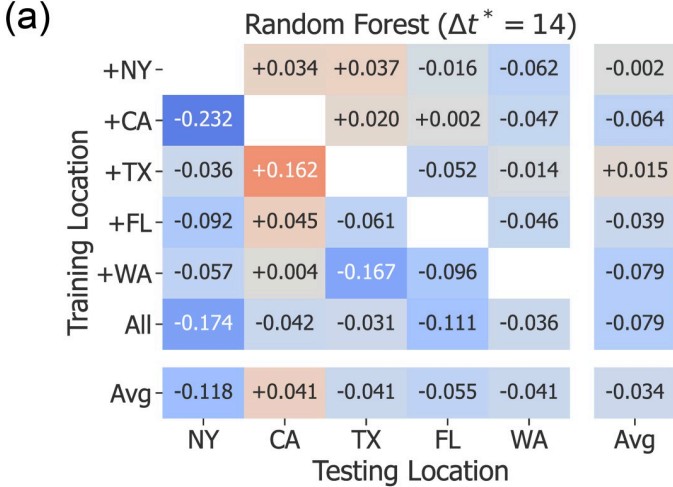

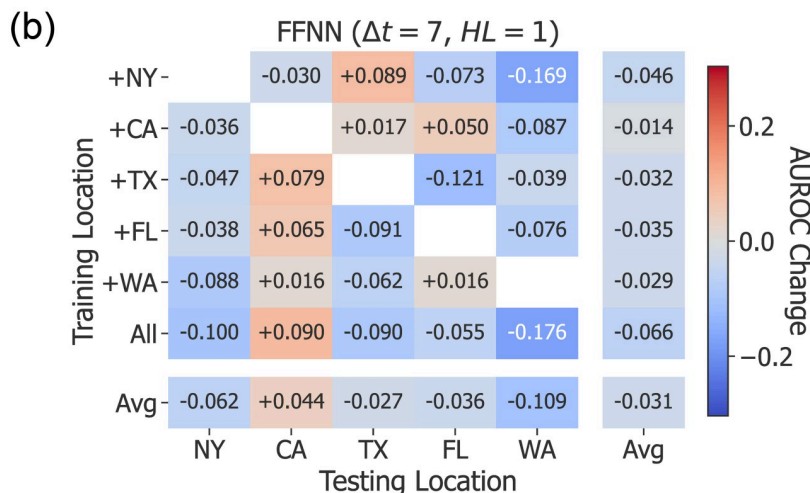

**Fig 9. Change in classification performance for models trained on multiple states.** Each cell in the heatmap represents a unique model, with states on the x-axis representing the testing location and states on the y-axis representing the state that supplemented the training data. The value in each cell is the change in AUROC compared to the baseline model. For example, the cell (NY, +CA) represents a model tested on New York data and trained on a combination of New York and California data, and the value −0.232 means that this model produced an AUROC 0.232 less than the baseline model that was only trained and tested on New York data (Table 3). Avg represents the average change in AUROC for the corresponding row or column.

surprising given our previous results, which have suggested that the importance of feature groups and the optimal observation windows vary between states, and further evidences the need for models to be tailored to individual states.

## Predictions on additional states

Until this point, we have limited our experiments and analysis to five states. While we also wanted to evaluate our model on other states, none of them experienced more than seven recorded attacks (Table 1). We found that with so few minority class examples it was impossible to learn an effective model. In an attempt to address this challenge, we considered the eight

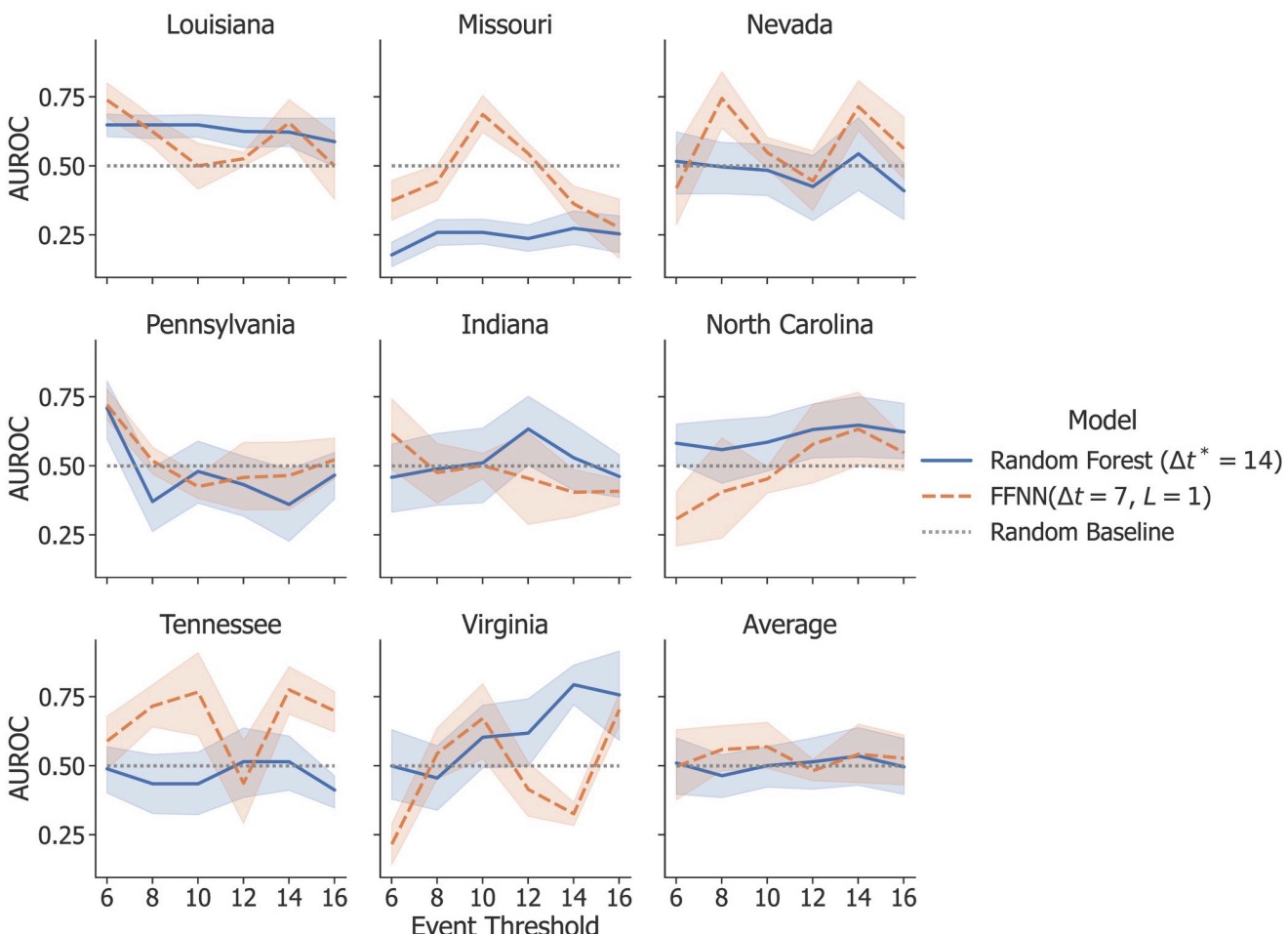

**Fig 10. Classification results on additional states.** The results on single-state testing show that neither RF nor FFNN perform significantly better than random on states that have experienced 5–7 attacks. Each plot shows AUROC as a function of the event threshold for each cross-validation experiment. The shaded areas represent the standard deviations of testing folds.

states that had experienced between five and seven attacks, and used hierarchical clustering (Euclidean distance, average link) to group them based on the observed news features. We tested two different grouping methods:

1. Single-state testing. After choosing a single state to evaluate and splitting the data into training and test dates, we used hierarchical clustering to supplement the training set with data from the states most similar to the testing state. We continued to add additional states to the training set until the number of attacks in the training set exceeded a fixed threshold. In order to maintain proper training and testing separation, we only supplemented the training set with records that matched the dates of the records in the original training set. Then we trained the model on the multi-state training set and evaluated it using the testing set from only the testing state.

2. Group testing. In these experiments, we also added data from the clustered states to the testing set, so that the states were evaluated as a group. For example, since Missouri (MO) is the state most similar to Louisiana (LA), we trained and tested the model on data from both MO and LA. The observed news features and number of attacks were very similar for these

**Table 4. Results for group testing on states that have experienced 5–7 attacks.**

| Testing State | Most Similar State(s) | Group Testing AUROC (5-fold cross-validation) | |
|---|---|---|---|
| | | Random Forest ($\Delta t^* = 14$) | FFNN ($\Delta t = 7, L = 1$) |
| LA | MO | .417 ± .149 | .605 ± .076 |
| MO | KS, LA | .466 ± .175 | .601 ± .097 |
| NV | UT, KY, MN | .459 ± .084 | .527 ± .031 |
| PA | OH, IL, VA | .500 ± .164 | .491 ± .164 |
| IN | OH, TN | .520 ± .110 | .575 ± .090 |
| NC | VA, MD, NJ | .548 ± .280 | .654 ± .214 |
| TN | KY, IN | .461 ± .108 | .419 ± .113 |
| VA | NC, MD, NJ | .548 ± .280 | .417 ± .151 |

The most similar state(s) are determined by hierarchical clustering on the news features and listed in order of similarity to the testing state.

states, which almost entirely eliminates the coarse evaluation problem we previously described.

Fig 10 presents the results from single-state testing using event thresholds between 6 and 16. Table 4 shows, for each testing state, the other states that were most similar and thus clustered, along with the results of group testing. The inconsistent and overall poor performance of both single-state testing and group testing is not surprising given our findings that supplementing the training data with examples from additional states reduced model performance, even though in this case we only included the most similar states. This suggests that in order to make meaningful predictions on these states, we either need more examples of attacks or a model that can account for more nuanced differences between states.

## Limitations

Key limitations of our study include the following:

1. Data availability. The window of overlap in data availability from GDELT and the GTD (1,413 days) is relatively small. This fact, coupled with other issues like the difficulty of the prediction problem and a noisy feature space, limits performance and introduces more variance to the modeling process. Further, since many states do not have a sufficient number of recorded attacks, the majority of our study is limited to five states.

2. Coarse location modeling. While we have shown that modeling terrorism prediction as a binary classification problem can be fruitful, our treatment of locations as entire states (on the basis of data availability) is a coarse abstraction of real-world locations. For example, New York is a large state and most of the attacks have taken place near New York City. Further, our methods do not account for geographical relationships, e.g. the fact New York City shares a large border with New Jersey (NJ). This limitation is strongly related to data availability.

3. Shifts in feature and class distributions over time, especially within news data. Although news data from GDELT has proven useful in this study, our framework assumes the distribution of these features will remain relatively stable over time. However, this may not be the case, as the sociopolitical landscape of any location can shift for a number of reasons beyond terrorist attacks.

## Conclusion

Terrorism presents a serious threat to global society. In this work, we used a series of machine learning models trained on localized news data to predict the occurrence of terrorist attacks in a given state and on a given day—a problem that, to the best of our knowledge, no prior work has attemped to solve. Our best model—a Random Forest that uses a novel Kolmogorov-Smirnov Moving Average method for feature representation—outperforms deep models and achieves an AUROC $\geq 0.667$ on four of five major states ($p \leq .0001$ as compared to random guessing) while being relatively insensitive to characterstics such as the amount of time between attacks or the type of attack. These results demonstrate that terrorist attacks can be predicted at a rate better than random, and show the potency of a multimodal approach in using news data to characterize a spatiotemporal location. Our results further indicate that the key factors limiting model performance are noise and the small number of attacks in the data set. Finally, our results affirm the need for localized models; in our case, manifested by creating separate models for each state. We envision a number of areas for future work, including:

1. Machine learning models that can better account for noise, either via manual feature engineering or noise-aware learning algorithms, and the differences between states.

2. The application of news data to other multimodal learning problems.

3. Continued efforts toward improving both the quality and quantity of terrorism-related data, including incorporating more sophisticated data augmentation strategies [46, 47].

4. Deeper studies into the relationship between news and specific terrorist attacks, such as investigating whether attacks can be traced to news topics like political events or socioeconomic themes.

Continued research in the use of machine learning to predict terrorist attacks promises to further our understanding of terrorism, inform strategies for mitigating or even preventing the attacks, and save lives.

## Supporting information

**S1 Appendix. Supplemental evaluation.** Includes the performance of other class imbalance strategies, runtime of K-S Moving Average, and analysis of the observation windows computed by K-S Moving Average.
(PDF)

**S1 Table. *p*-values for model results in Tables 2 and 3.**
(PDF)

## Author Contributions

**Conceptualization:** Steven J. Krieg, Christian W. Smith, Rusha Chatterjee, Nitesh V. Chawla.

**Data curation:** Steven J. Krieg.

**Formal analysis:** Steven J. Krieg, Christian W. Smith.

**Funding acquisition:** Christian W. Smith, Rusha Chatterjee, Nitesh V. Chawla.

**Investigation:** Steven J. Krieg, Christian W. Smith, Rusha Chatterjee.

**Methodology:** Steven J. Krieg, Christian W. Smith, Rusha Chatterjee.

**Project administration:** Rusha Chatterjee.

**Resources:** Nitesh V. Chawla.

**Software:** Steven J. Krieg.

**Supervision:** Nitesh V. Chawla.

**Validation:** Steven J. Krieg.

**Visualization:** Steven J. Krieg.

**Writing – original draft:** Steven J. Krieg.

**Writing – review & editing:** Steven J. Krieg, Christian W. Smith, Rusha Chatterjee.

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
