## [Decision Letter · Decision Letter 0]

23 Mar 2022

PONE-D-22-01703Predicting Terrorist Attacks in the United States using Localized News DataPLOS ONE

Dear Dr. Chawla,

Thank you for submitting your manuscript to PLOS ONE. After careful consideration, we feel that it has merit but does not fully meet PLOS ONE’s publication criteria as it currently stands. Therefore, we invite you to submit a revised version of the manuscript that addresses the points raised during the review process.

I request the authors to discuss about the limitations of the study presented. Problems related to Terrorist attacks prediction are required to be discussed. Recent references should be added to justify the proposed work.

We look forward to receiving your revised manuscript.

Kind regards,

Sathishkumar V E

Academic Editor

PLOS ONE

Journal Requirements:

[This material is based upon work supported by the Army Contracting Command - Aberdeen Proving Ground, Edgewood Division under Contract No. W911SR-19-C-0007. Any opinions, findings and conclusions or recommendations expressed in this material are those of the author(s) and do not necessarily reflect the views of the Army Contracting Command - Aberdeen Proving Ground, Edgewood Division. The authors declare no competing interests.]

 [This material is based upon work supported by the Army Contracting Command - Aberdeen Proving Ground, Edgewood Division under Contract No. W911SR-19-C-0007, which supported S.J.K., C.W.S., and R.C. The sponsors played no role in the study design, data collection and analysis, decision to publish, or preparation of the manuscript.]

4. We note that Figure 1 in your submission contain map images which may be copyrighted. All PLOS content is published under the Creative Commons Attribution License (CC BY 4.0), which means that the manuscript, images, and Supporting Information files will be freely available online, and any third party is permitted to access, download, copy, distribute, and use these materials in any way, even commercially, with proper attribution. For these reasons, we cannot publish previously copyrighted maps or satellite images created using proprietary data, such as Google software (Google Maps, Street View, and Earth). For more information, see our copyright guidelines: http://journals.plos.org/plosone/s/licenses-and-copyright.

a) You may seek permission from the original copyright holder of Figure 1 to publish the content specifically under the CC BY 4.0 license.  

Reviewers' comments:

Reviewer's Responses to Questions

**Comments to the Author**

1. Is the manuscript technically sound, and do the data support the conclusions?

Reviewer #1: Yes

Reviewer #2: Yes

Reviewer #3: Yes

Reviewer #4: Yes

2. Has the statistical analysis been performed appropriately and rigorously? 

Reviewer #1: No

Reviewer #2: Yes

Reviewer #3: Yes

Reviewer #4: Yes

3. Have the authors made all data underlying the findings in their manuscript fully available?

Reviewer #1: Yes

Reviewer #2: Yes

Reviewer #3: Yes

Reviewer #4: Yes

4. Is the manuscript presented in an intelligible fashion and written in standard English?

Reviewer #1: Yes

Reviewer #2: Yes

Reviewer #3: Yes

Reviewer #4: Yes

5. Review Comments to the Author

Reviewer #1: 1. The other classification methods like SVM, Decision tree are not used for comparison of results. Why?

2. Comparison with state-of-art approaches can be included

3. Statistical results to supplement the experiments can be dicussed

Reviewer #2: This paper presents a terrorist attack prediction in the US using local news data. I have no doubt to accept this paper for publication, however, some following remarks should be taken into consideration prior to publication:

- To handle imbalanced data, the authors have used a classic SMOTE technique. Please provide a justification why did the authors choose SMOTE? SMOTE has a major drawback such as it does not detect any noisy samples.

- Four machine learning algorithms were chosen in this study, however, these algorithms were very old. Please provide a justification concerning the selection of these algorithms since there are many algorithms available in the wild. In terms of SOTA, I think this paper should discuss XGBoost or other gradient-boosting algorithms.

Reviewer #3: In this work, the authors used a series of machine learning models trained on localized news data to predict the occurrence of terrorist attacks in a given state and on a given day—a problem that, no prior work has attempted to solve. The best model—a Random Forest that uses a novel Kolmogorov-Smirnov moving average for feature representation—outperforms deep models and achieves an AUROC ≥ 0.667 on four of five major states while being relatively insensitive to the amount of time between attacks. These results demonstrate that terrorist attacks can (to some degree) be predicted, and show the potency of a multimodal approach in using news data to characterize a spatiotemporal location. In general, this paper is well written and easy to follow. I would like to accept this paper if my following concerns are carefully addressed.

(1) The authors need to emphasise their contributions/novelties in the revision. In the current version, the authors did not discuss their contributions in detail.

(2) The proposed algorithm still can be improved if the ideas in the following papers are explored, i.e., "Making Sense of Spatio-Temporal Preserving Representations for EEG-Based Human Intention Recognition", "An Adaptive Semisupervised Feature Analysis for Video Semantic Recognition", and "A Semisupervised Recurrent Convolutional Attention Model for Human Activity Recognition". The authors are encouraged to discuss them in the revision.

(3) The authors should carefully proofread this paper and correct all the typos in the revision. In the current version, there are still some typos/grammar errors.

(4) Could the authors report the running time of the proposed algorithm? In this way, we can justify whether this algorithm can be applied to large-scale dataset.

Based on the above comments, I would like to accept this paper with major revision.

Reviewer #4: This paper used the machine leaning methods to predict the terrorist attacks with localized news data. It can predict whether a terrorist attack will occur on a given calendar date and in a given state. It is interesting and innovative. Following comments are given to future improve the quality.

1. It is suggested that the findings of this article do not need to be placed in the Introduction section.

2. In general, the socio-economic and geographical factors combined with the behavioral characteristics of terrorist organizations are used to predict the terrorist attacks. This paper adopts news data to build a feature model, which is very interesting. However, the accuracy of GDELT data is not high, which will may affect the results, please add the uncertainty analysis in the article.

3. In page6. line 531. You mentioned “If multiple attacks are carried out in a continuous period of time or location, they are recorded as a single incident.” How to build features of these continuous events? Is it the average of this period?

4. 208 unique location and date pairs were used to build the model. The author should justify if the hundred-level datasets are enough for an appropriate regression of the machine leaning methods used in this study. To my knowledge, the machine leaning is favorable for more data. But I’m sure for the RF and FFNN, the requirement of dataset number.

6. PLOS authors have the option to publish the peer review history of their article (what does this mean?). If published, this will include your full peer review and any attached files.

Reviewer #1: No

Reviewer #2: No

Reviewer #3: No

Reviewer #4: No

---

## [Decision Letter · Decision Letter 1]

16 Jun 2022

Predicting Terrorist Attacks in the United States using Localized News Data

PONE-D-22-01703R1

Dear Dr. Chawla,

We’re pleased to inform you that your manuscript has been judged scientifically suitable for publication and will be formally accepted for publication once it meets all outstanding technical requirements.

Kind regards,

Sathishkumar V E

Academic Editor

PLOS ONE

Additional Editor Comments (optional):

Reviewers' comments:

Reviewer's Responses to Questions

**Comments to the Author**

1. If the authors have adequately addressed your comments raised in a previous round of review and you feel that this manuscript is now acceptable for publication, you may indicate that here to bypass the “Comments to the Author” section, enter your conflict of interest statement in the “Confidential to Editor” section, and submit your "Accept" recommendation.

Reviewer #1: All comments have been addressed

Reviewer #4: All comments have been addressed

2. Is the manuscript technically sound, and do the data support the conclusions?

Reviewer #1: Yes

Reviewer #4: Yes

3. Has the statistical analysis been performed appropriately and rigorously? 

Reviewer #1: Yes

Reviewer #4: Yes

4. Have the authors made all data underlying the findings in their manuscript fully available?

Reviewer #1: No

Reviewer #4: Yes

5. Is the manuscript presented in an intelligible fashion and written in standard English?

Reviewer #1: Yes

Reviewer #4: Yes

6. Review Comments to the Author

Reviewer #1: All the suggestions given by the reviewers are addressed in the revised version of the manuscript and incorporate at the respective places.

Reviewer #4: (No Response)

7. PLOS authors have the option to publish the peer review history of their article (what does this mean?). If published, this will include your full peer review and any attached files.

Reviewer #1: No

Reviewer #4: No

---

## [Editor Report · Acceptance letter]

20 Jun 2022

PONE-D-22-01703R1 

Predicting Terrorist Attacks in the United States using Localized News Data 

Dear Dr. Chawla:

I'm pleased to inform you that your manuscript has been deemed suitable for publication in PLOS ONE. Congratulations! Your manuscript is now with our production department. 

Kind regards, 

on behalf of

Dr. Sathishkumar V E 

Academic Editor

PLOS ONE